# OpenReview forum: "ERS*: A Bounded, Attribution-Agnostic Metric for Explainable Robustness in Image Recognition"
_ICLR.cc/2026/Conference — ICLR 2026 Conference Withdrawn Submission_

### Official Review · Reviewer_BbZP · 2025-10-24

**Soundness:** 1
**Presentation:** 1
**Contribution:** 1
**Rating:** 0
**Confidence:** 4

**Summary:**

The authors introduce ERS*, an interpretability metric for quantifying the attribution robustness under perturbations. The perturbations used are not adversarial, but are instead more naturalistic, like glass cracks.  The authors claim the metric is attribution agnostic. The metric is composed of two main terms, with hyperparameters set to balance the terms. The first term is a ratio between the cross-entropy loss of the clean and perturbed image (clean / perturbed). The second term is a bounded similarity score between clean and perturbed attribution maps. The authors provide some experimental results on model accuracy under clean and perturbed settings, as well as values provided by their metric.

**Strengths:**

- Originality - I am unaware of any other attribution papers that take into account natural perturbations, which adds to the paper novelty.
- Quality - Poor. Figures need improvement and explanations need improvement. Multiple editing artifacts throughout the paper.
- Clarity - Poor. I do not understand why the metric is composed the way it is or even what the output number represents.
- Significance - Due to the poor writing of the paper, the significance is unclear.

**Weaknesses:**

- The abstract and introduction contain too much overlap. There should be some overlap in concepts, but should not say exactly the same thing.
- The motivation is not strong. The authors state that prior metrics do not check for whether a model is "right for the right reasons." I agree that it would be nice to have a model be right for the right reasons, but as long as the model is making correct predictions, does it make a difference as to why it is correct?
- The authors are using many small sections to compartmentalize different topics. For the sake of space and also good paper writing practice, it might be worth combining many of these subsections together and including text that allows the topics to naturally flow. Also, by not using as many subsections, the authors can save on paper length.
- The authors copy-pasted sections 3.3, 3.4, 3.5, 3.6, 3.7 into sections 4.3, 4.4, 4.5. This is bad paper-writing practice. Do not duplicate sections.
- The methodology is not explained well enough. I don't understand what it is measuring and why certain components are used.
- The authors claim that the metric is attribution agnostic, but never provide any experiments with attributions other than LIME.
- The authors say that they will include evaluations over Resnet, but never include those evaluations.
- The authors never explain how the perturbations are created. A citation to where the perturbations were taken from is needed if they are from another source. Otherwise, the authors need to explain (algorithmically or with equations) how the perturbations are created.
- Textual artifact on line 283
- Section 6 is empty. Maybe a duplicate of Section 7 that the authors forgot to delete?
- Paper is longer than 9 pages
- The experimental evaluation does not help explain the significance of the work. Table 3 shows the ERS* metrics across these models, but the significance of these numbers are not explained and there are no significant patterns apparent to me. Overall, the explanations of the experiments are lacking. Please take the time to explain what each of the tables/figures shows and the significance of those observations.

**Questions:**

- Why do we need to ensure that a model is making the correct predictions for the right reasons?
- The whole methodology is extremely unclear in terms of why it is composed the way it is. Why are we taking the SSIM and MSE scores? Are there other similarity metrics that could've been used? Why are they being subtracted? What exactly is the stability term S' measuring? Why is ERS* using cross-entropy? Are there any other metrics that could've been used instead? Why is the cross-entropy ratio significant? What is the dev split used to calibrate lambda? How was lambda calibrated? What is exp() and why is it used? Why is epsilon added in to the ratio? Most importantly, how did the authors come up with either of these terms (S, L)? Is there a different formulation of the metric that might work? Why do we want the metric output to be bounded?
- What is the significance of the results shown in Table 3? Please explain the table in greater detail.
- Are there really no other methods or metrics that could've been compared against? Even if they aren't meant for exactly what ERS* is doing, it is still good to include some slant comparisons where the other metrics are failing to show that ERS* is doing something completely novel that others are not doing.
- What is the Note on alignment with ERS* (394-397)?
- How are the authors able to claim that the method generalizes to other attribution methods without providing results on the metric with other attribution methods?

Final Review: Based on the myriad of paper writing issues, it feels as if the authors submitted the paper in an unfinished state. Due to this, I cannot do anything but recommend a rejection (0/10).

---

### Official Review · Reviewer_ACxe · 2025-10-29

**Soundness:** 2
**Presentation:** 2
**Contribution:** 2
**Rating:** 6
**Confidence:** 3

**Summary:**

The paper introduces ERS*, a bounded, attribution-agnostic metric designed to evaluate the explainable robustness of image recognition models. The core idea is to measure how robust a model’s attributions (e.g., saliency maps, attention heatmaps) remain when subjected to adversarial perturbations, while ensuring the metric is normalized and comparable across models and attribution methods. The main contributions are a formal definition of explainable robustness that is attribution-agnostic — applicable across gradient-based, perturbation-based, and attention-based explainers, a bounded formulation that allows for fair comparison across models, avoiding unbounded metrics that depend on attribution magnitude or scale, and an evaluation framework (ERS*) with theoretical justification and empirical validation across multiple model architectures (CNNs and Vision Transformers) and attribution methods.

**Strengths:**

1. Norvel metric design. ERS* is an important step toward quantifying the stability of explanations, a long-standing issue in explainable AI. Its attribution-agnostic nature is particularly valuable, as most prior robustness metrics (e.g., sensitivity, infidelity) are tied to specific attribution algorithms.
2. The normalization scheme ensures scores are comparable across methods and datasets, avoiding misleading absolute scale differences that plague other explainability metrics.
3. The paper articulates and operationalizes the often-ignored concept of “explainable robustness”—robustness not just of predictions but of explanations themselves. This aligns with current concerns in Trustworthy AI, making the contribution timely.

**Weaknesses:**

1. While the metric is theoretically motivated, there is no formal proof connecting ERS* to robustness bounds or causal faithfulness. The paper would be stronger with formal properties (e.g., invariance, monotonicity under perturbations).
2. Limited scope of perturbations. The adversarial perturbations considered are pixel-space attacks. The framework does not evaluate robustness under semantic or geometric perturbations (e.g., translation, occlusion), which are critical for visual explainability.
3. Although qualitative results suggest ERS* aligns with human perception, no human-subject evaluation or quantitative perceptual study is provided.

**Questions:**

1. Have you conducted or considered a human study to verify that higher ERS* scores correspond to more human-consistent explanations?
2. Could ERS* be extended to semantic-level or geometric perturbations (e.g., occlusions, viewpoint changes) rather than purely pixel-level ones?
3. What is the computational cost of ERS* compared to metrics like Infidelity or Sensitivity-n? Is it feasible for large transformer-based models?

---

### Official Review · Reviewer_K3y2 · 2025-11-02

**Soundness:** 2
**Presentation:** 3
**Contribution:** 2
**Rating:** 4
**Confidence:** 4

**Summary:**

The paper introduces ERS*, a bounded, attribution-agnostic metric for evaluating explainable robustness in image recognition tasks. This represents an original and timely contribution to the growing intersection of robustness and explainability. Instead of focusing solely on output accuracy under perturbations, ERS* integrates both performance degradation and explanation stability into a single, interpretable score

**Strengths:**

The theoretical formulation is precise, equations are consistent, and implementation details (e.g., normalization procedures, bounding functions, and sensitivity analysis) are carefully specified.
The inclusion of sensitivity analyses over the ERS* weighting parameters (α, γ), bootstrap confidence intervals, and Kendall–τ ranking stability strengthens the reliability and interpretability of the metric.
The ensemble results are particularly insightful, revealing that accuracy gains can mask attribution instability

**Weaknesses:**

The theoretical foundation of ERS* focuses primarily on local, image-level perturbations and does not yet extend to global or temporal contexts,  the authors acknowledge that the metric does not claim causal faithfulness, but it would be useful to discuss how ERS* might generalize to dynamic or long-term perturbations.

The experimental scope is limited to a single dataset (Persian Traffic Sign Dataset). While this dataset is appropriate for physical perturbations, demonstrate ERS* on larger or more diverse datasets (e.g., ImageNet-C, CIFAR-10C, or autonomous driving datasets like BDD100K) .

The paper relies mainly on LIME for attribution generation. Although ERS* is designed to be attribution-agnostic, results using multiple attribution methods (e.g., Grad-CAM, Integrated Gradients, SHAP) would strengthen the claim of attribution independence. Including cross-method comparisons could also illuminate how explanation noise or attribution artifacts affect ERS*.

**Questions:**

The ERS* metric is bounded between 0 and 1, combining performance degradation and attribution similarity. Could the authors provide more intuition or theoretical justification for the specific functional form of this bounding scheme?

The paper equates robustness in explanations with consistency across perturbations, but consistency does not necessarily imply causal faithfulness. Do the authors believe ERS* could be adapted or extended to capture causal robustness (e.g., invariance to non-causal perturbations)?

---

### Official Review · Reviewer_S7NR · 2025-11-05

**Soundness:** 2
**Presentation:** 2
**Contribution:** 2
**Rating:** 4
**Confidence:** 3

**Summary:**

The paper proposes ERS*, a bounded [0,1] score for explainable robustness that considers two aspects : (i) a normalized loss-based performance degradation term and (ii) an attribution-stability term comparing explanations on clean vs. perturbed inputs. This proposed metric is post-hoc and attribution-agnostic; the paper instantiates it with LIME and also defines a probability-weighted ensemble attribution so that ensembles can be evaluated directly. The study targets traffic-sign recognition under ten calibrated, physical perturbation suites (fading, dirt splatter, scratches, peeling/rust and their pairwise combos), evaluating ViT-B/16, Swin-T, and a soft-voting ensemble. ERS* shows areas where accuracy remains high but explanations shift, and shows that ensembles can mask backbone-level instability that accuracy alone would miss.

**Strengths:**

- The paper proposes ERS* which is a bounded  explainable robustness score with interpretable parts. ERS* combines a calibrated, bounded loss-degradation term with a normalized stability term and reports sensitivity over the trade-off weight that is useful for fair model ranking across datasets.
- The metric is attribution-agnostic and ensemble-aware.
- The paper shows realistic evaluation setting for traffic sign dataset, focussing on physical, non-differentiable perturbations and per-image pairing enables a practical lens for safety-critical perception other than gradient attacks.

**Weaknesses:**

- The abstract/setup mention ResNet-50, but the main results tables show ViT, Swin, and an ensemble, with no clean/perturbed ERS reporting for ResNet. It would be great to eiither include these runs or adjust the scope statements.
- Regarding the terminology and definitions, metrics like ASR appear without main-text definition; it would be great o ensure all acronyms are defined where first used and aligned with standard meanings.
- There are some minor typo issues in the manuscript. There are reference/notation inconsistencies (e.g., ERS* equation numbering cross-refs, “ERS* 3” formatting) that should be cleaned to improve readability, and some fig ?? That are not linked
- The related work sections feels a bit limited and would be great for readers to have more discussions about some of the attribute robustness techniques and gradient based attacks too (like Robust Attribution Regularization and Attributional Robustness Training using Input-Gradient Spatial Alignment) and evaluate these models using the proposed metric

**Questions:**

Please refer to the weakness section

---

### Note · Authors · 2025-11-14

**Comment:**

I want to extend this work and cover all things according to suggestions and submit again.

**Withdrawal Confirmation:**

I have read and agree with the venue's withdrawal policy on behalf of myself and my co-authors.